# Science through Wikipedia: A novel representation of open knowledge through co-citation networks

Wenceslao Arroyo-Machado[1,2], Daniel Torres-Salinas[1,2,3] *, Enrique Herrera-Viedma[4‡], Esteban Romero-Frías[1,5]

**1** Medialab UGR, University of Granada, Granada, Spain, **2** Department of Information and Communication, University of Granada, Faculty of Communication and Documentation, Granada, Spain, **3** EC3metrics spin off, University of Granada, Granada, Spain, **4** Andalusian Research Institute in Data Science and Computational Intelligence, University of Granada, Granada, Spain, **5** Department of Accountancy and Finance, University of Granada, Faculty of Economics and Business, Granada, Spain

☯ These authors contributed equally to this work.
‡ These authors also contributed equally to this work.
* torressalinas@ugr.es

**Data Availability Statement:** The data underlying the results presented in the study are available from Altmetric.com and Elsevier's CiteScore Metrics.

## Abstract

This study provides an overview of science from the Wikipedia perspective. A methodology has been established for the analysis of how Wikipedia editors regard science through their references to scientific papers. The method of co-citation has been adapted to this context in order to generate Pathfinder networks (PFNET) that highlight the most relevant scientific journals and categories, and their interactions in order to find out how scientific literature is consumed through this open encyclopaedia. In addition to this, their obsolescence has been studied through Price index. A total of 1 433 457 references available at Altmetric.com have been initially taken into account. After pre-processing and linking them to the data from Elsevier's CiteScore Metrics the sample was reduced to 847 512 references made by 193 802 Wikipedia articles to 598 746 scientific articles belonging to 14 149 journals indexed in Scopus. As highlighted results we found a significative presence of "Medicine" and "Biochemistry, Genetics and Molecular Biology" papers and that the most important journals are multidisciplinary in nature, suggesting also that high-impact factor journals were more likely to be cited. Furthermore, only 13.44% of Wikipedia citations are to Open Access journals.

## Introduction

Since its creation in 2001, Wikipedia has become the largest encyclopedic work human beings have ever created thanks to the collaborative, connected opportunities offered by the Web. Probably one of the most significant examples of Web 2.0 [1], Wikipedia represents a success story for collective intelligence [2]. With more than 170 editions, the English language version accounted for 5.5 million entries in January 2018 (approximately 11.7% of the entire encyclopedia). Given that worldwide Wikipedia is a top ten website in terms of traffic—according to Alexa (https://www.alexa.com/topsites, consulted on July 24, 2019)—and is one of the

**Funding:** This study has been possible thanks to financial support from "Knowmetrics: knowledge evaluation in digital society", a project funded by scientific research team grants from the BBVA Foundation, 2016, and the grant TIN2016-75850-R from the Spanish Ministry of Economy and Competitiveness with FEDER funds.

**Competing interests:** The authors have declared that no competing interests exist.

preferred results provided by search engines, it has become an outstanding tool for the dissemination of knowledge within a model based on openness and collaboration.

Perhaps Wikipedia's most important achievement has been to challenge traditional epistemologies based on authorship and authority and move towards a more social, distributed epistemology [3]. Wikipedia is therefore the result of a negotiation process that provides us with a representation of knowledge in society, offering tremendous research opportunities. For instance, some authors have studied the discursive constructions of concepts such as globalization [4] or historical landmarks like the 9/11 attacks [5]. The process of negotiation behind an article is often driven by the principles of verifiability and reliability in relation to the sources supporting the statements made. Specialized publications are among the preferred sources of reference (https://en.wikipedia.org/wiki/Wikipedia:Identifying_reliable_sources, consulted on July 24, 2019), mainly in the form of scholarly material and prioritizing academic and peer-reviewed publications, as well as scholarly monographs and textbooks.

Consequently, the social construction of knowledge on Wikipedia is explicitly and intentionally connected to scholarly research published under the peer-review model. This has offered us the opportunity to investigate how Science and Wikipedia interrelate. Although Wikipedia is not a primary source of information, some studies have examined citations of Wikipedia articles [6,7]. Moreover, numerous studies have analyzed how Wikipedia articles cite scholarly publications because contributors are strongly recommended to do so by the encyclopedia itself. Studies have focused on the analysis of reference and citation patterns in specific areas of knowledge [8], on exploring Wikipedia's value as a source when evaluating scientific activity [9], or on Wikipedia's role as a platform that promotes open access research [10].

Furthermore, some studies undertaken within the last decade could be said to be framed within the Altmetric perspective because they have used indicators extracted from the social media to measure dimensions of academic impact [11,12,13]. Wikipedia references to scientific articles can provide highly valuable altmetric information given that the inclusion of references is not a trivial activity and is usually subject to community scrutiny. For instance, the Altmetric Attention Score—an indicator created by Altmetric.com—gives this type of citation a high value (3) that is higher than mentions on Facebook (0.25) or Twitter (1), but lower than references to blogs (5) and news feeds (8).

Networks have also been used for knowledge representation in order to visualize differences between the Universal Decimal Classification category structure and that generated by Wikipedia itself [14], to generate automated taxonomies and visualizations of scientific fields [15], and to show connections between articles [16]; furthermore, studies based on the complex networks approach have also been reported [17]. One way to address knowledge representation from a bibliometric perspective is through the use of co-citations [18], an approach that uses references in common received from a third document as a proxy for similarity between two scientific documents. Co-citations have been used to observe similarities between words [19] or areas of knowledge [20].

From an Altmetric perspective, the concept of co-citation was transferred to the online world giving rise to co-link analysis [21], where documents are replaced by webpages or websites, and citations are replaced by links. Co-link analysis has successfully mapped scientific knowledge [22] and analyzed fields such as universities [23], politics [24] or business [25].

These different concepts where recently combined and applied to Wikipedia by Torres-Salinas, Romero-Frías and Arroyo-Machado [26] and tested in the field of the Humanities by mapping specialties and journals. The present study uses Wikipedia to draw a social representation of scientific knowledge and the areas into which it is divided. After collecting all the references in Wikipedia, we concluded that only 5.49% correspond to the Humanities (see S1

Table). Therefore, in the present study we take the same approach in investigating Science as a whole, including the Humanities. We seek to achieve the following objectives:

- To apply co-citation analysis to all the articles referenced in Wikipedia in order to validate their usefulness in analyzing open knowledge platforms;

- To offer a general portrait of the use of scientific literature published in journals through the analysis of references and their obsolescence. Thus, we hope to be able to describe the consumption of scientific information by the Wikipedia community and detect possible differences between fields; and lastly, as the nuclear objective of our paper

- To discover the different visions offered by Wikipedia by using co-citation networks at different levels of aggregation: 1) journal co-citation maps 2) main field co-citation maps 3) field co-citation maps. Through these representations we intended to obtain a holistic view of how scientific articles in Wikipedia are used and consumed.

## Materials and methods

### Information sources and data pre-processing

The main source of information in this study was Altmetric.com, one of the most important platforms gathering altmetric data about scientific papers. The total volume of references to scientific papers made by Wikipedia articles was downloaded on April 11, 2018. This amounted to 1 433 457 references published between October 15, 2004 and April 10, 2018, citing 960 017 discrete resources. Initially we pre-processed the data with R in order to clean it up. This involved correcting errors to facilitate links with other data sources, eliminating duplicate references, and deleting references lacking the data needed for our study, such as publication dates. As a result, the total number of references fell to 1 211 904 citing 857 087 individual resources. ISSNs corresponding to these resources were collected using the Altmetric API. A total of 36 090 ISSNs corresponding to 693 805 scientific articles were obtained. In addition, we used Elsevier's CiteScore Metrics (with data updated to February 6, 2018) to link each scientific article to its source through journal identifiers and thus obtain additional information. The references were linked to Elsevier's CiteScore Metrics's entire collection. Fig 1 summarizes this process.

The Scopus ASJC (All Science Journal Classification) offered in the CiteScore Metrics collection has been used to attribute areas, main fields, and fields to the scientific articles being studied. To use Scopus (https://service.elsevier.com/app/answers/detail/a_id/12007/supporthub/scopus/) terminology, we would say that the ASJC identifies four major areas each of which includes several *Subject Area Classifications* (termed *main fields* in our study). Given that *multidisciplinarity* is a common main field in each of the four areas, we have decided to include this category as a main area as well. As a result, there are 27 main fields (*subject area classifications* in Scopus terminology) and 330 fields (*fields*) within five main areas (*subject areas*): namely "Health Sciences", "Life Sciences", "Physical Sciences", "Social Sciences & Humanities", and "Multidisciplinary". Hence, the final sample consists of 847 512 references included in 193 802 Wikipedia entries, citing 598 746 individual scientific articles from 14 149 journals. This process of attribution enabled us to identify references to scientific articles and, at a more aggregated level, references to journals, fields and main fields, giving rise to three different co-citation networks.

### Statistical analysis

As part of the descriptive statistics, the mean, median, standard deviation and interquartile range have been calculated for the number of references made by Wikipedia and the citations

**Fig 1. Methodological process of collecting the massive dataset of papers referenced in Wikipedia and assigning them to different scientific categories.**

received by the scientific articles, as well as for the dates of citation and publication of the papers, at all the different levels under study. We would emphasize the fact that in our dataset all Wikipedia entries include at least one reference to scientific papers and all articles and journals included have been cited at least once by Wikipedia articles. Furthermore, the obsolescence of the scientific references has also been calculated using the Price index [27], which has been applied to intervals of up to 5, 10, 15 and 20 years, the entire dataset, and by scientific fields. The Price index refers to the percentage of publications cited not older than a specific number of years and is a means of showing the level of immediacy of publications cited, which differs according to the scientific area [28]. Similarly, the distributions of citations between Wikipedia and Scopus have been compared using the citation value recorded by Elsevier's CiteScore Metrics—, which corresponds to the sum of citations in 2016 to articles published between 2013 and 2015—, and by Wikipedia, and adjusted to allow for this limitation. Finally, the distribution of journal citations from Wikipedia has been analyzed, to determine whether it fits power law and log-normal distributions using the poweRlaw package [29].

## Analysis of co-citation networks

Co-citation networks, bibliographic coupling and direct citations are some of the most significant bibliometric networks we can use to map citations from Wikipedia entries; of these, co-citation networks are the most popular in research [30,31]. If we take into account other types of network such as co-author and co-word, the aforementioned three methods show a high degree of similarity [32]. Within the field of altmetrics, the concepts of co-citation and coupling have both been adapted [33], but co-citations offer more varied alternatives [34]. Furthermore, they are of special interest as they have been identified as capable of enhancing transdisciplinarity [35]. Hence, we have generated co-citation maps at the level of journal, field and main field.

The Pathfinder algorithm [36] has been applied as a pruning method following a common configuration (r = ∞, q = n-1) that reduces the networks to a minimum covering tree. This algorithm—successfully applied in the field of Library and Information Sciences [26, 37]—keeps only the strongest co-citation links between all pairs of nodes and offers a diaphanous view of large networks. Given the huge amount of co-citations, especially between journals, we use this technique to prune them in order to make the networks more explanatory. Since it is applied to values in relation to distances, the inverse value of the co-cites has been used in our analysis. Local measures of proximity, betweenness and eigenvector centrality have also been calculated. In the case of journals, the data has undergone a second pruning to eliminate those entries with a co-citation degree lower than 50.

## Results

### General description

We have analyzed 847 512 references to scientific articles distributed across 193 802 Wikipedia entries. A total of 598 746 scientific articles published in 14 149 journals are cited. Each Wikipedia entry includes 4.373 (± 8.351) references to scientific articles, while they receive a mean of 1.415 (± 10.15) citations. Some 81.71% of the total number of scientific articles (489 235) receive only one citation and this corresponds to 57.73% of all references in the study sample.

This high standard deviation can be explained by looking at the top 1% of Wikipedia entries with more references, some 60.874 references (± 32.752), representing 13.92% of all references in the study. This top 1% of entries is related to listings—highlights of scientific events in a discipline during a given year—history, genes, common diseases or drugs and medicines. For instance, the highest number of references recorded for a single entry is 550 (https://en.wikipedia.org/wiki/2017_in_paleontology). Furthermore, the level of variation in the standard deviation is not unfamiliar in metrics of this type since the distribution studied here has an especially marked asymmetry because 81% of papers receive only one citation and 97% of the total only receive between one and three. Moreover, 20% of the most cited papers only receive 40% of the total number of citations. This phenomenon occurs in almost all bibliometric indicators [38].

Analyzing the evolution of Wikipedia citations over time, we find that in 2009 the number of individual articles cited and references in Wikipedia fell with respect to 2007 and 2008. However, since then constant growth has been observed (Fig 2). If we take as a reference the first citation year per Wikipedia entry, in its first year each entry receives 2.793 citations (63.871% of all references), falling to 0.341 (7.795%) and 0.249 (5.682%) in the second and third years, respectively, and further decreasing year after year. Hence, old entries do not accumulate more citations and—except those referenced in 2007 (an average of 6.448) and 2008 (4.022)—these amount to between 2 and 3 per year.

The mean publication date of the scientific papers cited is 2001; most were published between 1988 and 2018 (88.51%) as Fig 3 shows. Some 39.43% were published between 2008 and 2018. To analyze the literature on obsolescence, we used the Price index [27], which reflects the percentage of references within a given period. Our results indicate that 36.84% of citations appear within 5 years of publication, twice as many appear within 15 years, and 83.46% appear within 20 years (Fig 4).

A total of 549 201 papers (91.72%) are co-cited through Wikipedia references and give rise to 7 810 091 co-citations with an average of 28.442 (± 77.088) per paper. To better understand this huge variation, the two most co-cited papers are https://www.altmetric.com/details/3201729 (4997 mentions) and https://www.altmetric.com/details/3216022 (3591 mentions) with 3559 co-citations. From the total number of co-citations, 6 110 250 (78.24%) establish

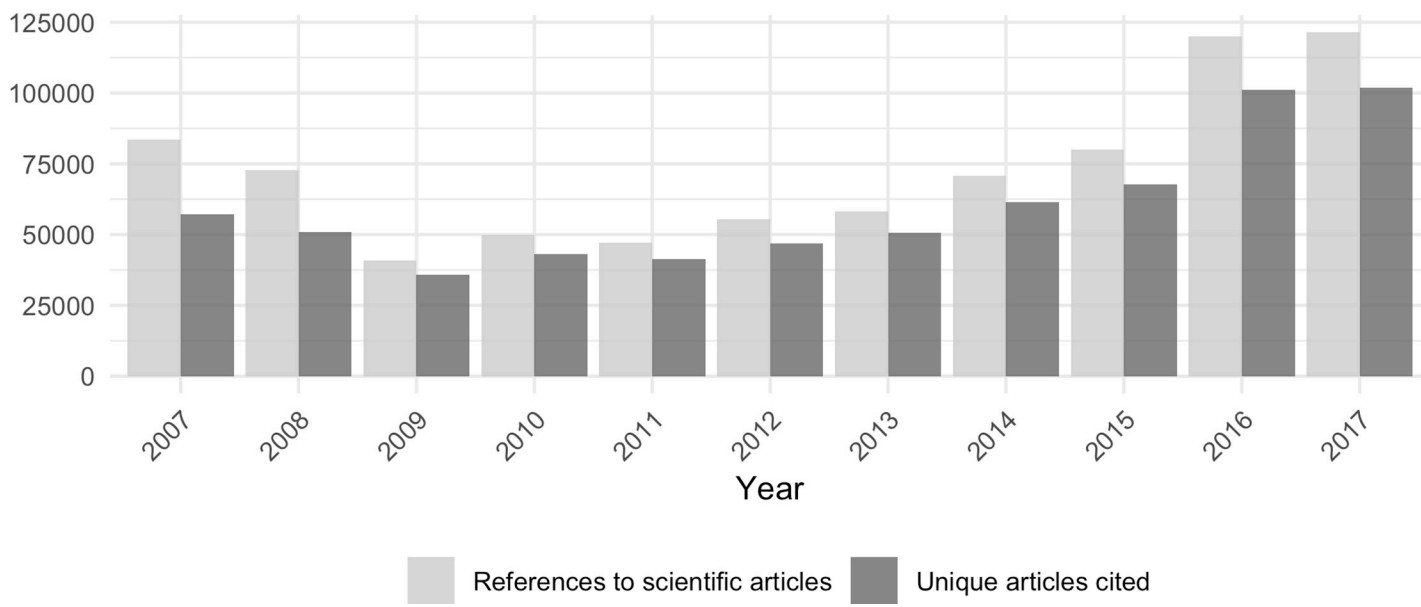

**Fig 2. Annual values of total references made by Wikipedia and single articles cited.**

connections between papers in different main fields and 1 699 841 (21.76%) do so with papers in the same field. Multidisciplinary co-citations are also slightly more broadly distributed as they have an average of 1.626 (± 3.513) co-citations, compared to non-multidisciplinary co-citations 1.045 (± 1.021).

We have studied the distributions of Wikipedia and Scopus citations at the journal level, considering in both sets only those made in 2016 to articles published between 2013 and 2015. The relationship between the two has been analyzed using linear ($R^2 = 0.486$) and generalized additive models ($R^2 = 0.572$)—quantile-quantile (Q-Q) plot shows that both distributions are highly skewed to the right (See S1 Fig)—. As can be seen in the scatter plot (Fig 5) and log-log scatter plot (See S2 Fig), several journals stand out in both metrics: *PLoS One*, *Nature*, Science

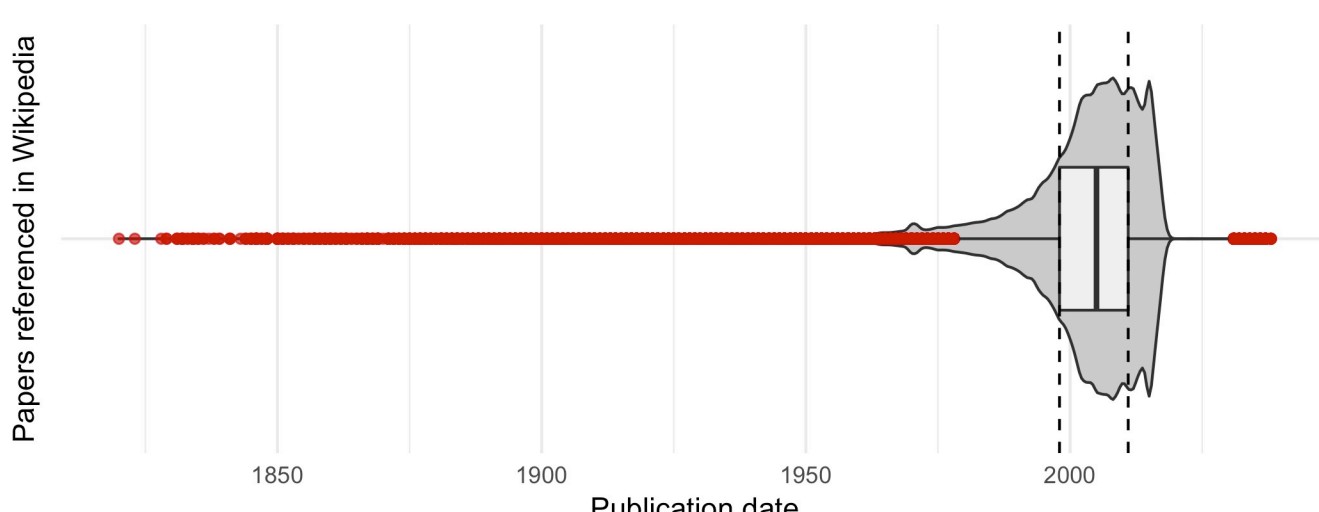

**Fig 3. Box and violin plots for the years of publication of the scientific articles referenced in Wikipedia (outliers are shown in red).**

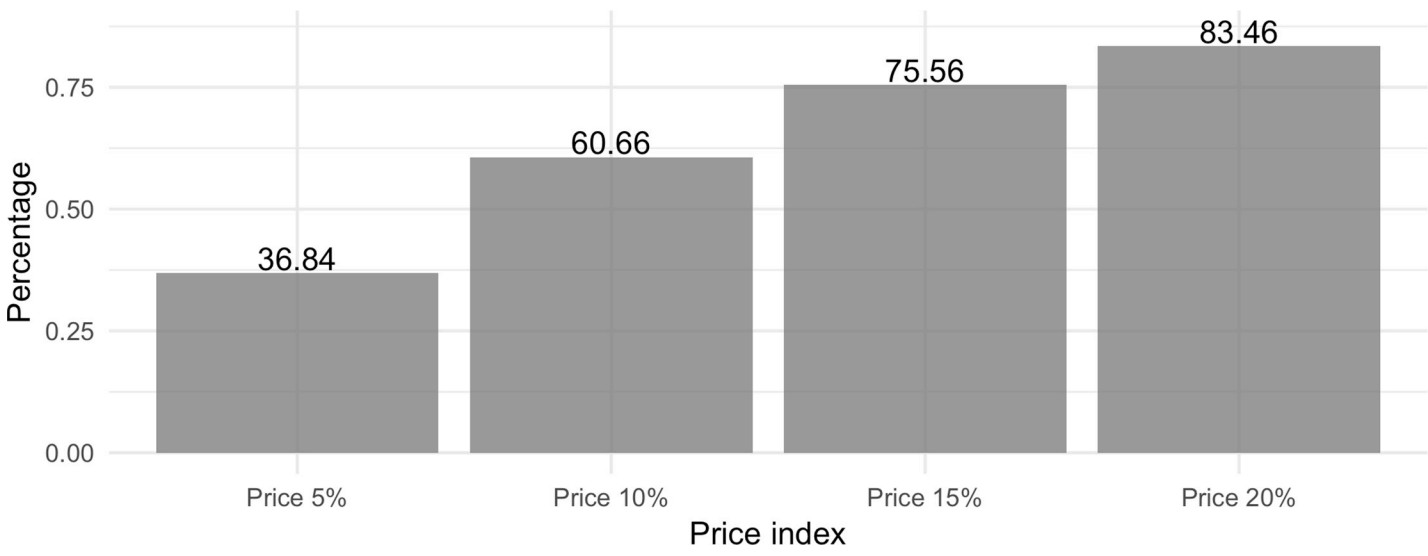

**Fig 4. Literature obsolescence of Wikipedia article references using the Price index for 5, 10, 15 and 20 years.**

and *Proceedings of the National Academy of Sciences of the United States of America (PNAS)*. In this sense we have obtained the journals' citation percentiles in Wikipedia and Scopus, using only journals with a minimum of three citations in both platforms and two articles cited to avoid noise, and then the ratio between these percentiles have been calculated. While the commented journals have the same attention (ratio = 1), the over-cited ones in Wikipedia are *Mammalian Species* (3559), *Art Journal* (192.56), *Northern History* (126.92), *European Journal of Taxonomy* (83.92) and *Art Bulletin* (80.92), and the under-cited ones are *Physical Review A—Atomic, Molecular, and Optical Physics*, *Dalton Transactions* and *Applied Surface Science* (all of them with 0.00027). Furthermore, the distribution of total Wikipedia citations follows a power law, obtaining a p-value of 0.29 through the goodness-of-fit test, using a bootstrapping procedure. Power law and log-normal distributions offer acceptable fits to the data and do not differ (See S3 Fig), giving a p-value of 0.971 via Vuong's test.

To illustrate these differences, we have analyzed the 20 most cited scientific articles in Wikipedia (see S2 Table). 14 are related to biology (mostly genetics-oriented), while the rest are related to astronomy, physics and computer science, although they also focus on astronomy-related topics. When comparing the Wikipedia citations of these articles (mean 1223.1, ± 1167.19) with the Scopus database (534.1, ± 716.07), we found a mean absolute difference of 1000.4 citations (± 965.42). Only four of these articles received more citations in Scopus than in Wikipedia. The most cited article in Wikipedia is "Generation and initial analysis of more than 15,000 full-length human and mouse cDNA sequences" with 4997 citations (compared to 1228 in Scopus).

### Journals by areas

The 14 149 journals in our sample have a mean 42.36 (± 269.22) articles cited in Wikipedia, with each journal receiving a mean 59.9 (± 458.54) citations. Wikipedia entries include a mean 3.25 (± 4.82) references to different scientific journals. So, there are five areas and each journal belongs to one or more of them with 3279 in "Social Sciences & Humanities", 3077 in "Health Sciences", 2489 in "Physical Sciences", 1298 in "Life Sciences" and 31 in "Multidisciplinary", while the rest belong to more than one area.

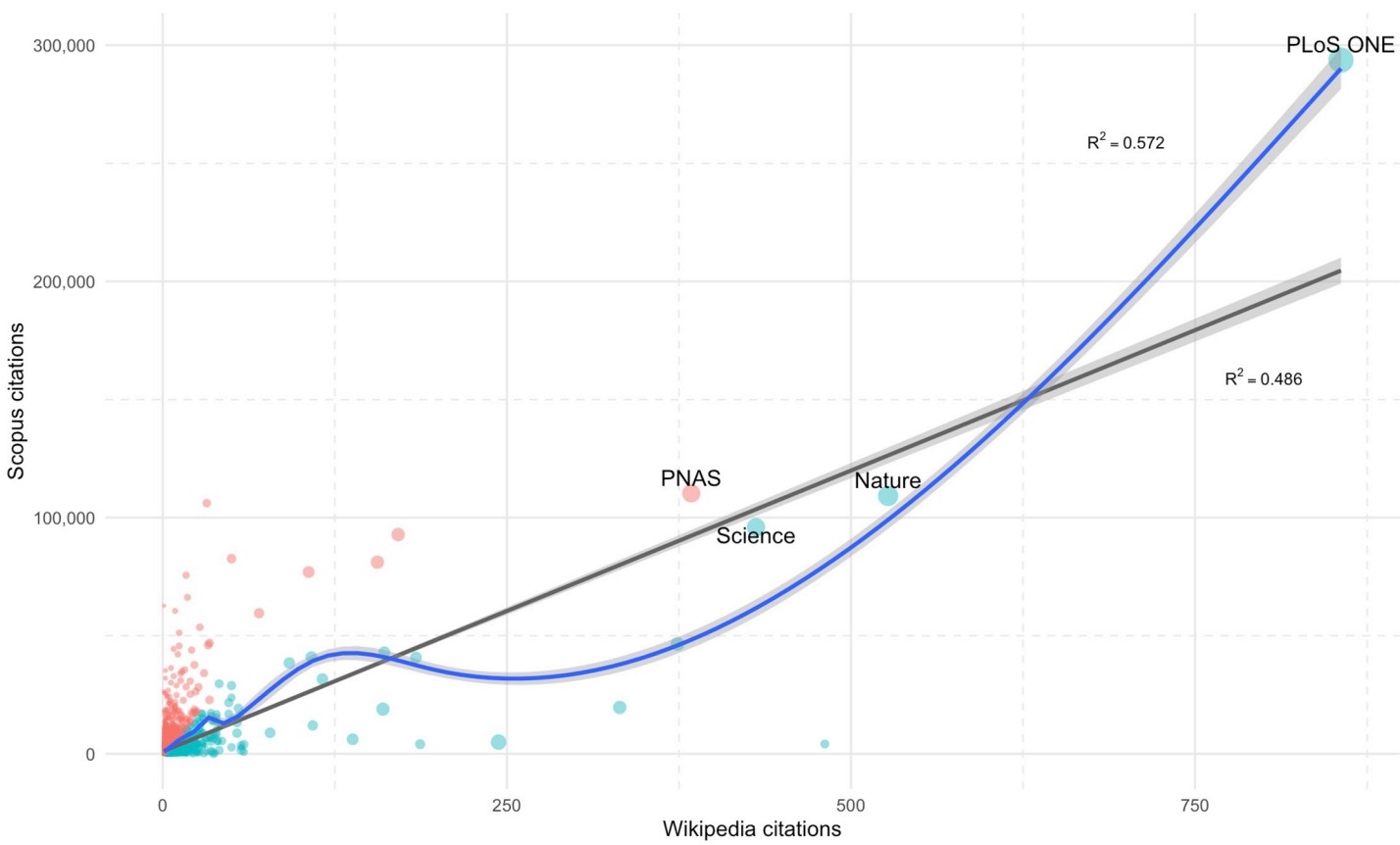

**Fig 5. Scatter plot of journals by citation collected in Scopus and Wikipedia in 2016 to articles published between 2013 and 2015.** The size of the points corresponds to the number of articles published in that period and the color corresponds to the ratio between citation percentiles: red (more on Scopus) and blue (more on Wikipedia).

The most cited journals are *Nature* (26 434 citations); *PNAS* (24 104); and the *Journal of Biological Chemistry* (21 921), which also has the highest number of individual articles cited (16 611). What is remarkable is the fact that only 13.44% of citations are to Open Access journals, when Wikipedia explicitly supports free content. Only two of the 20 most cited journals (see S3 Table) are open access resources (*PLoS One* and *Nucleic Acids Research*).

Our map of co-citation networks between journals reveals that 13 474 journals (95.2% of the total) are co-cited–each journal has an average of 10.165 co-citations (± 28.292)–, with only 30 (0.22%) having no relationship with the main component (Fig 6). This giant component is made up of 1 156 668 relationships (Fig 6A), but when we apply the Pathfinder algorithm it is reduced to 684 473 (Fig 6B). While the first figure shows that *Science* is the most important journal with the highest number of co-citations (7119) and the highest betweenness, proximity and eigenvector centrality scores (see S4 Table), second comes *PNAS* (1604). By analyzing the co-citations between journals by areas in the global network, we find a similar proportion of them are co-cited with others from the same area and from a different one in "Health Sciences" (47.64%, 52.36%), "Physical Sciences" (50.71%, 49.29%) and "Social Sciences & Humanities" (53.93%, 46.07%), but there are significant differences in "Life Sciences" (31.04%, 68.96%). "Multidisciplinary" (0.66%, 99.34%) shows the highest contrast but consists of only a few journals.

However, after applying the Pathfinder algorithm (S4 Table), which eliminates the weakest co-citation links between a journal and co-cited journals, the network obtained is also pruned to display only nodes with a minimum of 50 co-cites. So the score for *Science* falls to 33, below

**A**

**B**

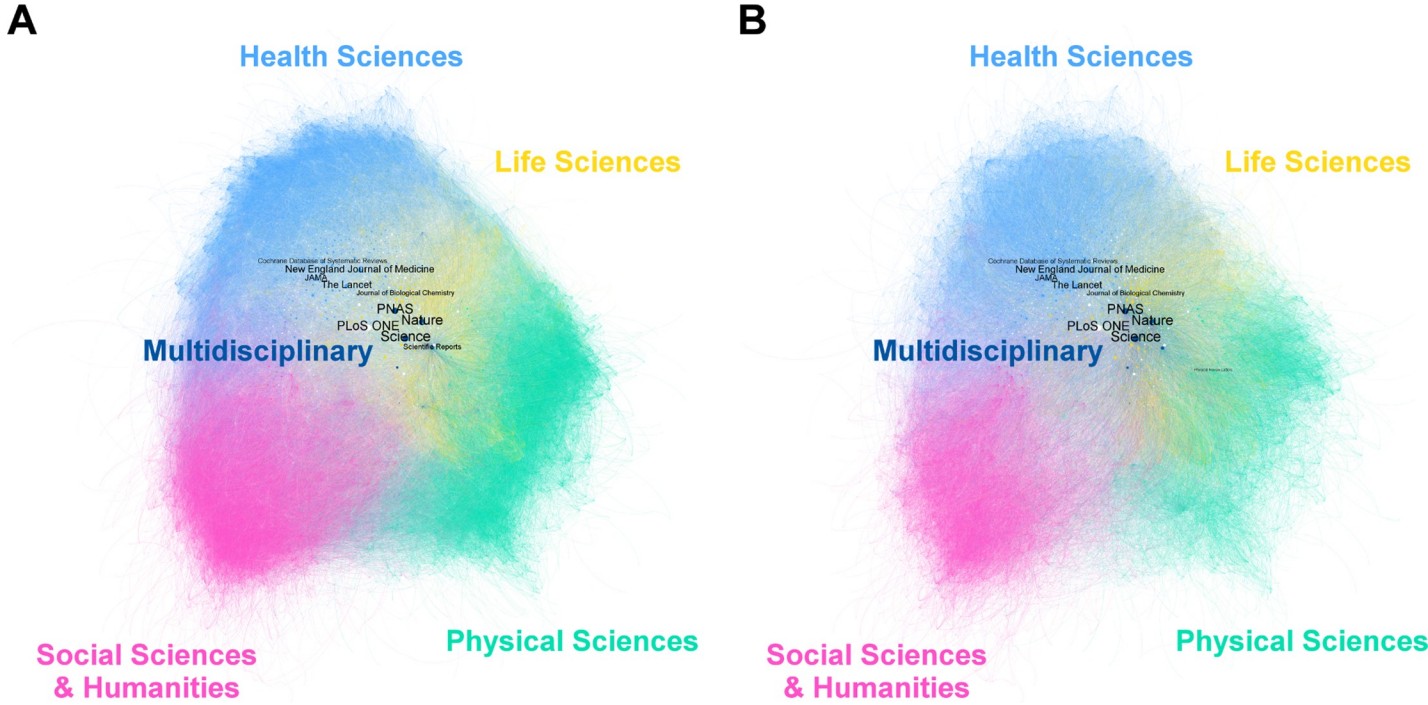

**Fig 6. Co-citation network of journals based on Wikipedia article references.** A) Main component of the full network; B) Pathfinder of the full network. Each node represents one journal and node size corresponds to the total number of citations received; color corresponds to the area but those with more than one are white; the thickness of the edges corresponds to the degree of co-citation between the two. The titles of the 10 journals with the highest intermediation value have been included.

*PNAS* (251), *Nature* (76) and the *Journal of Biological Chemistry* (41). Fig 7 shows the network resulting from applying the Pathfinder algorithm, based on a minimum of 50 co-cites.

If we look at the journals' areas of knowledge we find that Scopus distinguishes between four main subject areas ("Physical Sciences", "Health Sciences", "Social Sciences" and "Life Sciences") and one transversal area called "Multidisciplinary". As S5 Table shows, "Life Sciences" is the most frequently referenced area in Wikipedia (414 400 references and 4.03 mean references per entry) whereas "Multidisciplinary" has the highest average citation (1.88). Given that some journals can be attributed to more than one of the four areas, additional areas have been generated as a result of the possible existing combinations for viewing journals on the net. The network in Fig 5 shows that most journals belong to "Life Sciences" (36.6% of the total), followed by "Life Sciences & Health Sciences" (19.2%), "Health Sciences" (14.5%) and "Physical Sciences" (14.5%). "Social Sciences & Humanities" is in sixth position (3.5%) and "Multidisciplinary" is eighth (1.1%). *PNAS*, *Nature* and *Science* not only act as major intermediaries in the network but also show their multidisciplinary nature by reflecting very strong co-citations with journals from different fields. This is particularly notable in both *Nature* and *Science*. Most connections linked with *PNAS* are to journals in "Life Sciences" and "Health Sciences & Life Sciences". *PLoS ONE* also shows strong co-citation links with journals in many areas despite being cataloged in "Health Sciences & Life Sciences".

## Main fields

Wikipedia entries that reference articles within the same main field do so with an average of 1.466 (± 1.504) references, while entries that mix articles from different main fields do so with 5.764 (± 9.799).

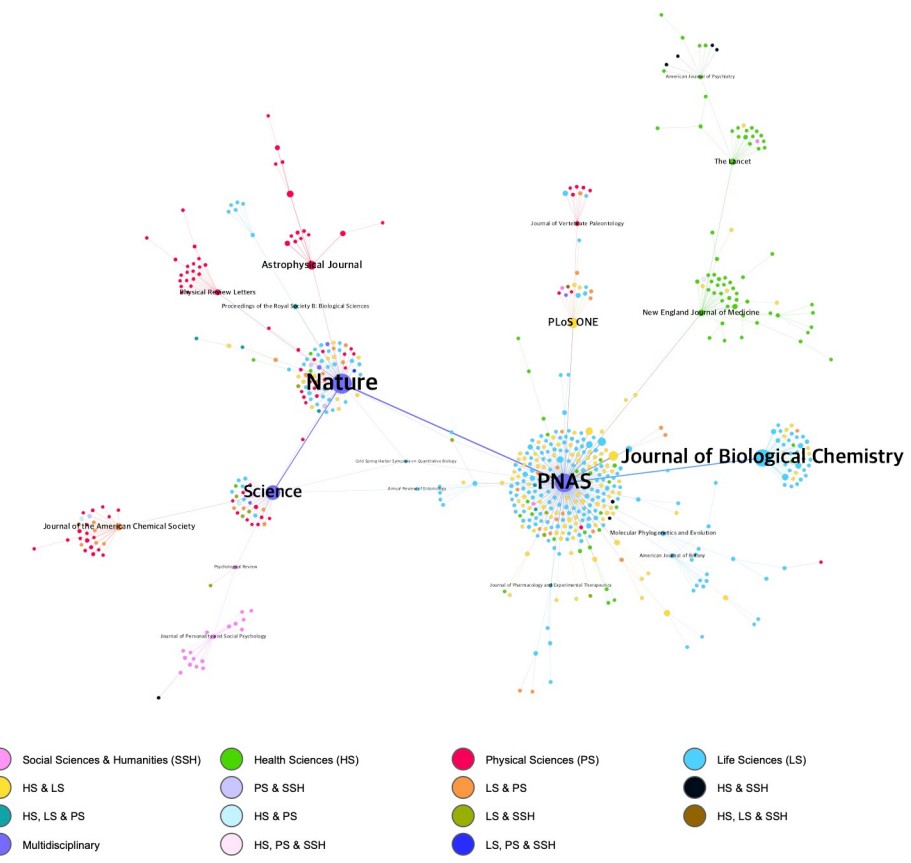

**Fig 7. Co-citation network of journals based on Wikipedia article references.** This network is produced by applying the Pathfinder algorithm—based on a minimum of 50 co-cites—and shows a total of 629 relationships. Each node represents one journal and node size corresponds to the total number of citations received; color corresponds to the area or combination of subject areas to which it belongs; and the thickness of the edges corresponds to the degree of co-citation between the two. The titles of the 20 journals with the highest intermediation value have been included.

Within the 27 main fields (see S1 Table), "Medicine" (referenced in 72 384 Wikipedia entries; 3.81 mean references per entry) and "Biochemistry, Genetics and Molecular Biology" (referenced in 64 945 Wikipedia entries; 4.11 mean references per entry) are the most significant. In contrast, "Dentistry" has the lowest level of presence in Wikipedia entries (992), although the mean number of references is 2.42. The main fields with the lowest means are: "Arts and Humanities" (1.65) and "Decision Sciences" (1.6).

In relation to citations received by main fields from scientific papers (see S1 Table), in absolute terms articles in "Medicine" (206 576 citations received) and "Biochemistry, Genetics and Molecular Biology" (181 954) stand out. However, on average, the outstanding main fields are "Multidisciplinary" (1.88 citations per article) and "Earth and Planetary Sciences" (1.88). "Dentistry" remains the least frequently cited area and has the lowest mean number of citations (1.14).

Fig 8 shows the distribution by main field of all articles included by Scopus, a total of 62 821 260 scientific articles indexed in the database, by comparison with the distribution by main field of articles cited in Wikipedia (see S6 Table). The main fields attributed to the articles correspond to those of the journals in which they are published. Note that from the Wikipedia perspective, there is a greater presence of articles from "Biochemistry, Genetics and Molecular Biology" (10.86% more), "Agricultural and Biological Sciences" (4.72% more), "Multidisciplinary" (4.37% more), "Earth and Planetary Sciences" (2.11% more), "Immunology and

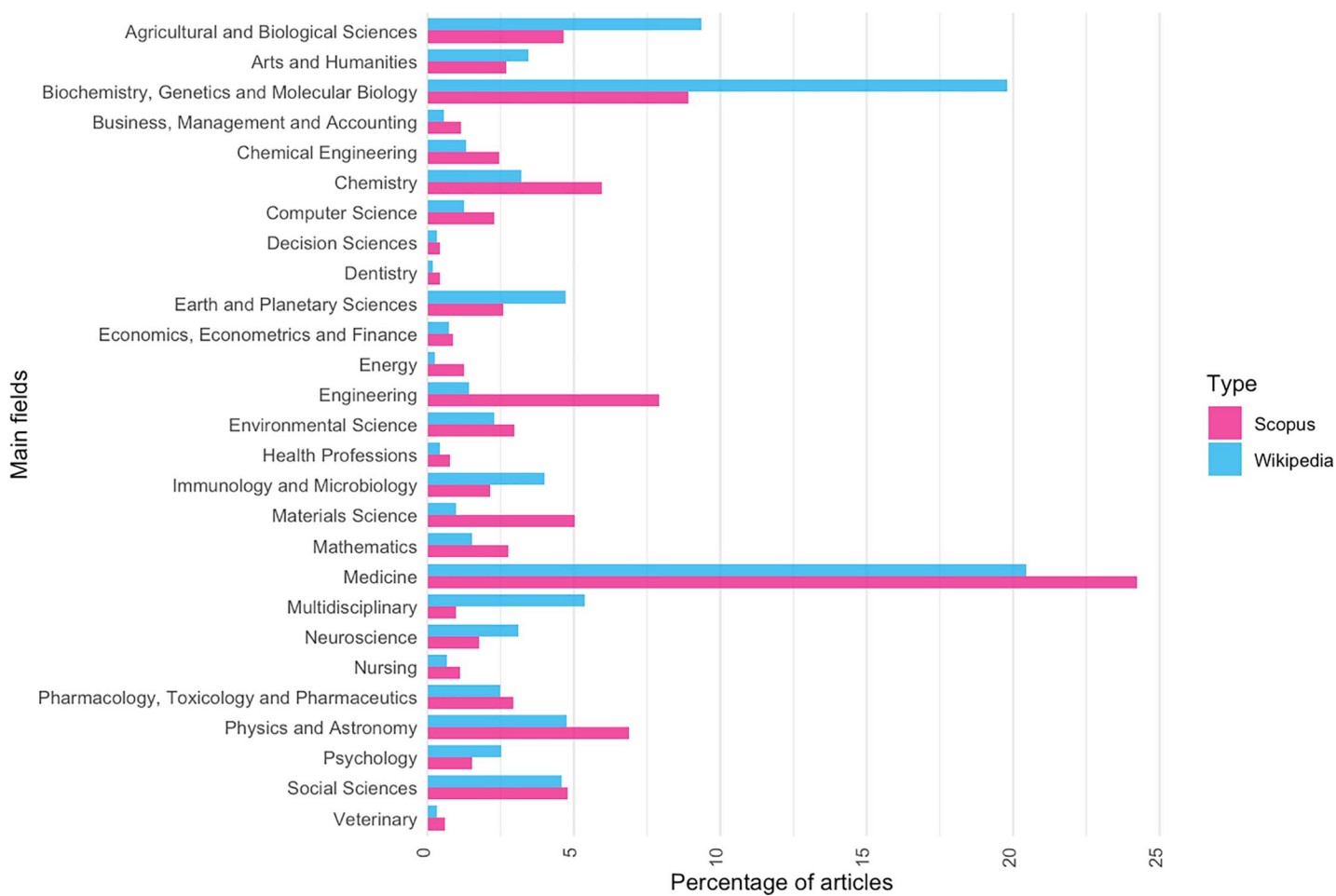

**Fig 8. Comparison of the percentage of articles by main field in Scopus and Wikipedia.**

Microbiology" (1.88% more), and "Neuroscience" (1.34% more) than that found in Scopus. In contrast, in Scopus the proportion of articles from "Engineering" (6.49% more), "Materials Science" (4.05% more), "Medicine" (3.76% more), Chemistry (2.72% more) and "Physics and Astronomy" (2.13% more) is higher than that in Wikipedia. The main fields for which the distribution of articles is similar both in Scopus and Wikipedia are: "Social Sciences"; "Economics, Econometrics and Finance"; "Decision Science" (with differences of less than 0.2% in absolute terms).

Analysis of the Price index for each of these main fields shows that "Energy" and "Material Sciences" reflect a rather limited degree of obsolescence compared to the rest (See Fig 9). This phenomenon is more noticeable in the former, with a value of 55% for the first five years, reaching 91.56% when we extend the time interval to 20 years. "Arts and Humanities" and "Decision Sciences" are in a very different situation, with Price indexes for the first five years of 22.76% and 24.67%, respectively—half that of "Energy" for the same period. When we look at Price indexes for 20 years, we also see considerably lower values with 68.44% in "Arts and Humanities" and 60.54% in "Decision Sciences", the latter also having the lowest value of all main fields over 20 years.

Fig 10 shows the co-citation network of the 27 main fields after applying the Pathfinder algorithm. The two main actors are "Medicine" and "Biochemistry, Genetics and Molecular

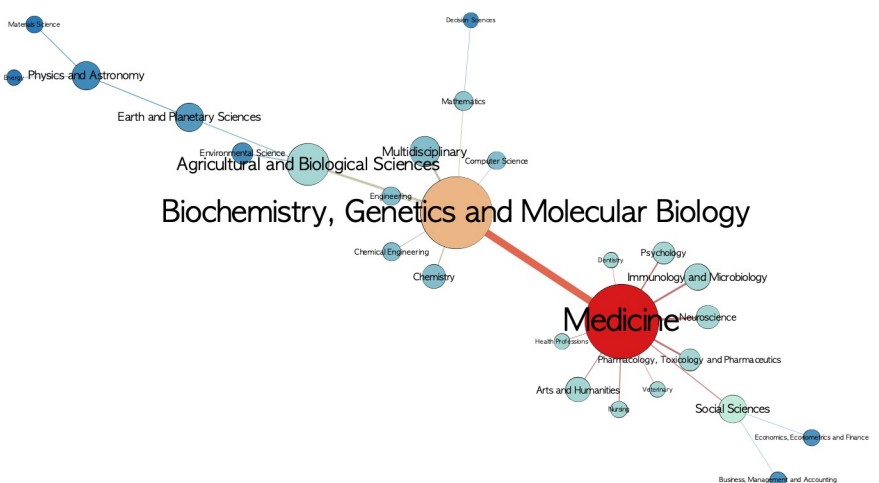

**Fig 9. Price index for Wikipedia main fields.**

Biology", which constitute the core of the network and share strong co-citation links. Apart from the connection between "Medicine" and main fields linked to Health, the strong relationship with "Arts and Humanities" and "Social Sciences" (also as a link between "Business,

**Fig 10. Co-citation network of the 27 main fields after applying the Pathfinder algorithm.** The nodes represent each main field; node size corresponds to the total number of citations received, color corresponds to own vector centrality; and the thickness of the edges corresponds with degree of co-citation.

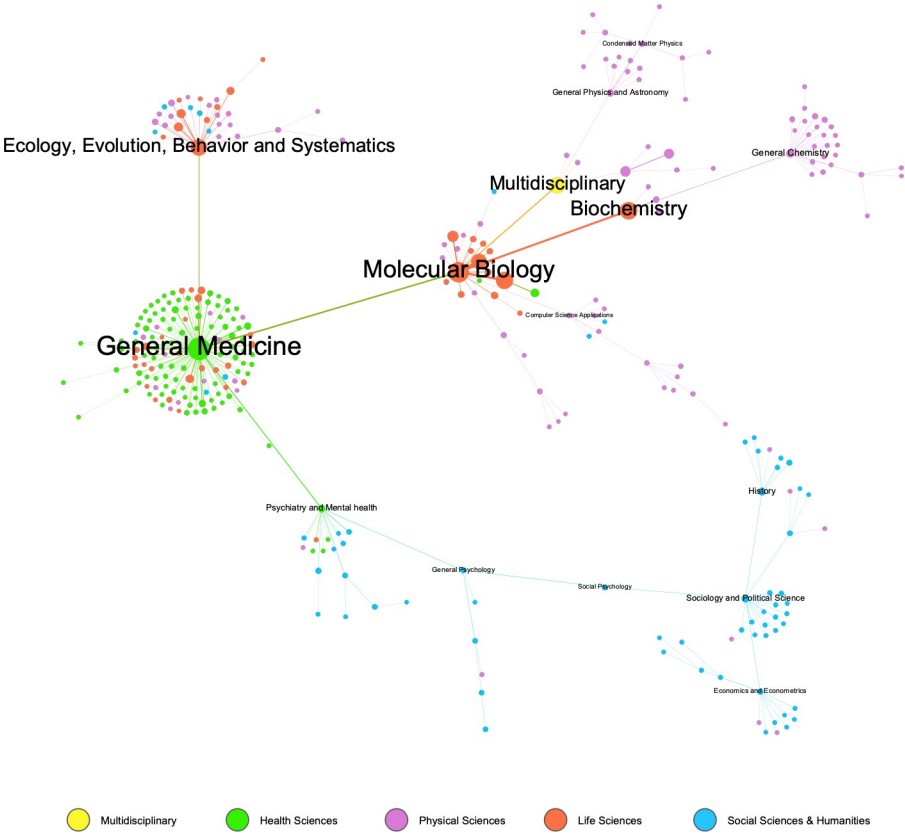

**Fig 11. Co-citation network of the 330 fields after applying the Pathfinder algorithm.** The nodes represent each field, indicating size, total number of citations received, color, thematic area or areas, and the thickness of the edges indicates the degree of co-citation. Field titles are given for the 15 fields with the highest levels of intermediation.

Management and Accounting" and "Economics, Econometrics and Finance") stands out. Furthermore, it is also noteworthy that "Biochemistry, Genetics and Molecular Biology" is closely linked to "Agricultural and Biological Sciences", highlighting the connections with more tangential main fields such as "Computer Science", "Engineering", "Multidisciplinary" and "Mathematics".

### Field

As can be seen in S7 Table, within the 330 fields studied, the two most outstanding fields are: "General Medicine" and "Molecular Biology", with 108 131 and 98 118 total citations, respectively. Both stand at a considerable distance from the next outstanding specialties: "Biochemistry" (78 704), "Genetics" (77 920) and "Multidisciplinary" (72 346).

Fig 11 shows the co-citation network of the 330 after applying the Pathfinder algorithm. This network shows the prominent position of "General Medicine", which has the highest number of relevant co-citations and is central to the majority of fields in "Health Sciences". We should also mention the role of "General and Social Psychology" as a connection between "Health Sciences & Humanities" and "Social Sciences". Despite the link to "Social Psychology", the "Social Sciences & Humanities" appear disconnected and are structured around three fields: "Sociology and Political Science", "Economics" and "Econometrics and History". Finally, the fields related to Physics also appear in peripheral areas of the graph and are linked to "Multidisciplinary"; "General Chemistry", which is linked to "Biochemistry", is in a similar situation.

## Discussion

We have conducted a large-scale application of co-citation analysis to all articles referenced in Wikipedia. Previous research [26] had experimented with this approach in the Humanities alone, presenting promising results in mapping Science from the Wikipedia perspective. However, it was necessary to validate this on a more ambitious scale, that is, the entire open online encyclopedia, in which the Humanities represent only 5.49% of all the references collected. The results presented above are indicative of how this this innovative approach allows us to depict a complete picture of Science.

Thus we can produce science maps that complement the traditional co-citation maps focused on scientific articles [39, 40] and provide a representation of knowledge that focuses on the vision and use of information in the scientific community. The methodology presented, which focuses on the co-citation of Wikipedia articles, offers holistic maps of the use of scientific information by Wikipedia users/editors who are not necessarily scientists. Therefore these maps represent the user's vision of scientific activity and in this sense they are close to other mapping methodologies that are not exclusively centered on citations but centered on the user —maps such as those based on Clickstream Data [41], readership network maps using Mendeley [42] or maps based on Co-Tweet [34]. By comparison with earlier research, the main novelties of the present study are that for the first time a source of information as important as Wikipedia has been used, several sources have been combined (Altmetrics, Scopus), and we have used Pathfinder, which is a much more efficient algorithm.

The Wikipedia references, unlike those collected in other social media, offer remarkable quality control and transparency. In relation to the problem of trolling, the encyclopedia is based on a solid quality management system of post-publication peer review in which, in the case of discrepancies, changes are resolved through consensus between editors. Wikipedia also has two manuals: for non-academic experts (https://en.wikipedia.org/wiki/Help:Wikipedia_editing_for_non-academic_experts) and for researchers, scholars, and academics (https://en.wikipedia.org/wiki/Help:Wikipedia_editing_for_researchers,_scholars,_and_academics) both specifying the importance of the use of citations under the principles of verifiability and notability. This substantially minimizes the likelihood that references in entries will be tampered with.

Wikipedia also offers a complete list of its bots (https://en.wikipedia.org/wiki/Special:ListUsers/bot), including those such as the Citation bot (https://en.wikipedia.org/wiki/User:Citation_bot), which in addition to adding missing identifiers to references, corrects and completes them, something for which the digital encyclopedia offers several tools (https://en.wikipedia.org/wiki/Help:Citation_tools). However, this does not prevent the appearance of publications with a high, anomalous number of citations [43]. For instance, we found a report (https://www.altmetric.com/details/3171944) cited in 1450 lunar crater entries, not attributable to a bot. So, although the use of citations is not compromised, practices of this sort must be taken into account, for example, if their use is in an evaluative context. Given all of the aforementioned, we consider that in this context in which 193 802 Wikipedia entries and 847 512 article citations have been analyzed, it is very difficult to produce manipulations that could significantly alter the system and, consequently, the results achieved here.

### About the results

This study illustrates the use of scientific information from the Wikipedia perspective, which is the most important and largest encyclopedia available nowadays. We have been able to determine the main fields that receive citations in Wikipedia entries. The most relevant fields are "Medicine" (32.58%), "Biochemistry, Genetics and Molecular Biology" (31.5%) and

"Agricultural and Biological Sciences" (14.91%). In contrast, "Dentistry" (0.28%), "Energy" (0.43%), "Decision Sciences" (0.49%) and "Veterinary" (0.52%) are the main fields that globally receive fewer references. We would emphasize the fact that these areas need to strengthen the visibility of their work. In general, we find it remarkable that Science disciplines should dominate the Humanities and Social Sciences.

If we look at the maps at journal level, we find that the most important publications are multidisciplinary in nature and the main journals in terms of centrality are *Science*, *Nature*, *PNAS*, *PLoS ONE*, and *The Lancet*. However, after applying the Pathfinder algorithm to discard less relevant relationships, we note that *PNAS* rises to first position, limiting the centrality of journals like *Science* and *PLoS ONE*, which have more but weaker co-citation relations. Without a doubt, this places *Science* and *PLoS ONE* in an interesting centrality space and turns them into nodes that connect with a curiously wide variety of journals. Likewise, our proposed methodology has enabled us to detect the strongest links between the main journals and their scientific uniqueness; in this sense, it is worth highlighting *Nature*'s strong relationship with "Physics", *Science*'s relationship with "Chemistry" and that of *PNAS* with "Life Sciences".

Like other platforms, multidisciplinary journals are hubs and articulate the Wikipedia network. However, despite being a common global phenomenon, Wikipedia does have, and contains unique citation practices mentioning journals that are not cited or mentioned in a relevant way in other databases or platforms. This is evidenced in the scatter plot of Wikipedia and Scopus citations (see Fig 5).

This difference is also illustrated by a comparison of the journals most mentioned in Altmetric.com with those most mentioned in Wikipedia. As we can see, Wikipedia has the major multidisciplinary journals in common, as does Scopus. However, some of the most frequently mentioned journals in Wikipedia are the *Journal of Biological Chemistry* or *Zookeys* which are located in JCR's Q2. Nonetheless, the *Journal of Biological Chemistry*, for example, is one of the most widely cited journals in the field of genetics receiving a large number of citations in various entries such as "Androgen receptor" (45 citations) or "Epidermal growth factor receptor" (25 citations). Therefore, Wikipedia points to another type of specialized journal in different fields (See S8 Table) that are not identified in other rankings. In addition, as we have indicated, this can hardly be due to trolling or a bot.

If we observe the map of main fields, "Medicine" and "Biochemistry, Genetics and Molecular Biology", are the two main nodes. From this perspective, "General Medicine" is the most relevant node, accounting for the highest number of citations received. It acts as a highly important connector in the network. Moreover, this underlines the role of "Psychology" in connecting "Health Sciences" with "Social Sciences and the Humanities".

Given the open nature of Wikipedia, the analysis of references to open access journals is particularly relevant. Firstly, it is remarkable that only 13.44% of citations are to Open Access journals, when Wikipedia explicitly supports free content. Furthermore, only two of the 20 most cited journals are open access resources (*PLoS ONE* and *Nucleic Acids Research*). Teplitskiy et al. [10] determined that the odds in favor of an Open Access journal being referenced in the encyclopedia were about 47% higher than that of closed access journals. They also suggested that high-impact factor journals were more likely to be cited, as we have also observed in our results. In relation to open access resources, the fact that many articles in closed journals can be accessed through their authors or third parties [44] may distort some of these considerations.

*PLoS ONE* is the most relevant open access journal cited in Wikipedia. And it is the fourth in terms of centrality, just behind *Science*, *Nature* and *PNAS*, all three of which operate under a non-open access model. When applying Pathfinder, *PLoS ONE*'s centrality is reduced. This is due to the fact that this method eliminates the weakest co-citation links, which are highly

relevant to the journal because although is cataloged in "Health Sciences & Life Sciences", it occupies a central position in the network in relation to journals from vastly different areas.

Our study has also shown that certain fields have a stronger relative presence in Wikipedia references than in Scopus. This is the case of "Biochemistry, Genetics and Molecular Biology" (10.86% more), "Agricultural and Biological Sciences" (4.72% more) and "Multidisciplinary" (4.37% more), among others. This could indicate that from the Wikipedia perspective some fields receive more attention than from the scientific community as a whole. Finally, in relation to obsolescence we have observed significant differences between main fields. For instance, for the first five years, "Energy" has 55% of references, whereas the "Arts and Humanities" receives only 22.76%.

## Comparison of Wikipedia and Scopus

Wikipedia's view of science differs from that of Scopus. While linear regression and generalized additive models have a correlation statistically significant, we do not establish causality due to the high presence of outliers that do not obey these patterns. Also, the focus of thematic attention provided by Wikipedia editors shows striking differences, an aspect that is clearly evident in Scopus and Wikipedia's differences of coverage and the presence in the latter of journals in very prominent positions that do not coincide with the views of other altmetrics sources. At the level of article citations the strong asymmetry in the distribution curve of Wikipedia citations is due to the fact that most receive only between one and three citations, showing a much more extreme phenomenon than Pareto's law. However, at the journal level, we can confirm the existence of a power-law distribution that shows a phenomenon similar to that observed in citations in Scopus [45]. This is why these differences allow us to appreciate that we are dealing with two phenomena that are not the same.

## Limitations

As in our previous study [26], some limitations derive from the attribution of categories to journals since journals do not always belong to the category assigned by the database [46]. Latent co-citation can also arise [37] because some journals may be assigned to more than one field. We have resolved this issue by combining all of them under the same label.

It should be noted that the methodology used, which combines various sources (Altmetric. com, Wikipedia and Journal Metrics by Elsevier), generates certain limitations. For example, we have only taken account of scientific articles since only resources with an ISSN and indexed by Scopus have been used; this excludes books or chapters of special relevance in the Humanities [47]. This is an issue present in the classical approaches that are limited to scientific journals [48].

Other problems derive from the sometimes inaccurate Altmetric description of their records. The dataset frequently presents duplicate records; errors in the year of publication, DOI and identifiers assigned to records; or records with many fields containing null values. For instance, the field presenting most problems has been that of the ISSN, which is sometimes incorrect in both Altmetric and CiteScore Metrics.

One of the most striking limitations detected with regard to the use of Wikipedia references as a measure of activity impact lies in their volatility because many references can be created or eliminated very quickly, making data collection and subsequent use difficult. In this respect Altmetric.com's Altmetric Attention Score can also mislead because given that it is a static measure, it only takes account of presence and makes no allowance for frequency. However, none of these limitations affects the overall results of our study because the large number of references and processed articles in our sample minimizes their impact.

Finally, we must point out that these types of map are an interesting complement to quantitative information offered by platforms such as Altmetric.com or PlumX. Thus, thanks to these contextual methodologies [49] it is possible to elucidate more clearly the social impact (societal impact) of scientific articles in particular and of Science in general of platforms such as Wikipedia. In the future we will extrapolate co-citation studies to other documentary typologies and platforms included in Altmetric.com such as news or policy reports in order to clearly establish the different representations of knowledge generated by different users and consumers of scientific information.

## Supporting information

**S1 Table. Descriptive statistics of references made by Wikipedia entries and citations that scientific articles receive from Wikipedia entries by main fields.**
(PDF)

**S2 Table. Most cited articles in Wikipedia.**
(PDF)

**S3 Table. Most cited journals in Wikipedia.**
(PDF)

**S4 Table. Main journals by measures of centrality in full co-citation network, co-citation PFNET and filtered co-citation PFNET.**
(PDF)

**S5 Table. Descriptive statistics of references made by Wikipedia entries and citations that articles receive from Wikipedia by areas.**
(PDF)

**S6 Table. Percentage of articles by main field in Scopus and Wikipedia and its differences.**
(PDF)

**S7 Table. Top 25 most cited fields with local measures of centrality.**
(PDF)

**S8 Table. Comparison of the most cited journals on Wikipedia and Altmetric.com.**
(PDF)

**S1 Fig. Normal and log-normal quantile–quantile plots of journal citations collected in Scopus and Wikipedia in 2016 to articles published between 2013 and 2015.**
(TIFF)

**S2 Fig. Log-log scatter plot of journals by citation collected in Scopus and Wikipedia in 2016 to articles published between 2013 and 2015.**
(TIFF)

**S3 Fig. Power law and log-normal fits in cumulative distribution of journal citations from Wikipedia.**
(TIFF)

## Acknowledgments

We thank Altmetric.com for the transfer of the data that has allowed us to conduct this study.

## Author Contributions

**Conceptualization:** Daniel Torres-Salinas, Enrique Herrera-Viedma, Esteban Romero-Frías.

**Data curation:** Wenceslao Arroyo-Machado.

**Funding acquisition:** Esteban Romero-Frías.

**Methodology:** Daniel Torres-Salinas.

**Project administration:** Esteban Romero-Frías.

**Software:** Wenceslao Arroyo-Machado.

**Supervision:** Enrique Herrera-Viedma.

**Visualization:** Wenceslao Arroyo-Machado.

**Writing – original draft:** Wenceslao Arroyo-Machado, Daniel Torres-Salinas, Esteban Romero-Frías.

**Writing – review & editing:** Daniel Torres-Salinas, Esteban Romero-Frías.

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
