## [Decision Letter · Decision Letter 0]

24 Oct 2019

PONE-D-19-25486

Science through Wikipedia: a novel representation of open knowledge through co-citation networks

PLOS ONE

Dear Dr. Torres-Salinas,

Thank you for submitting your manuscript to PLOS ONE. After careful consideration, we feel that it has merit but does not fully meet PLOS ONE’s publication criteria as it currently stands. Therefore, we invite you to submit a revised version of the manuscript that addresses the points raised during the review process.

As you can see below, both reviewers found your work interesting but also raised concerns about several aspects of it. Your revision should especially address:

1.- Comments by reviewer 1 about the lack of clear, specific research questions.

2.- Several methodological comments made by both authors (please, notice PLOS ONE publication criterion #3, https://journals.plos.org/plosone/s/criteria-for-publication#loc-3).

3.- The need to further develop the interpretation and discussion about the obtained results (in line with the journal's publication criterion #4, https://journals.plos.org/plosone/s/criteria-for-publication#loc-4).

We would appreciate receiving your revised manuscript by Dec 08 2019 11:59PM. To enhance the reproducibility of your results, we recommend that if applicable you deposit your laboratory protocols in protocols.io, where a protocol can be assigned its own identifier (DOI) such that it can be cited independently in the future. For instructions see: http://journals.plos.org/plosone/s/submission-guidelines#loc-laboratory-protocols

We look forward to receiving your revised manuscript.

Kind regards,

Sergi Lozano

Academic Editor

PLOS ONE

**Journal Requirements:**

2. We note that some of the Figures in your submission contain copyrighted images. All PLOS content is published under the Creative Commons Attribution License (CC BY 4.0), which means that the manuscript, images, and Supporting Information files will be freely available online, and any third party is permitted to access, download, copy, distribute, and use these materials in any way, even commercially, with proper attribution. For more information, see our copyright guidelines: http://journals.plos.org/plosone/s/licenses-and-copyright.

a) You may seek permission from the original copyright holder of Figure(s) [#] to publish the content specifically under the CC BY 4.0 license.

3. Thank you for stating that “The funders had no role in study design, data collection and analysis, decision to publish, or preparation of the manuscript” in your financial disclosure.

Please also provide the name of the funders of this study (as well as grant numbers if available) in your financial disclosure statement.

**Comments to the Author**

1. Is the manuscript technically sound, and do the data support the conclusions?

Reviewer #1: Partly

Reviewer #2: Partly

2. Has the statistical analysis been performed appropriately and rigorously? 

Reviewer #1: No

Reviewer #2: Yes

3. Have the authors made all data underlying the findings in their manuscript fully available?

Reviewer #1: Yes

Reviewer #2: Yes

4. Is the manuscript presented in an intelligible fashion and written in standard English?

Reviewer #1: Yes

Reviewer #2: Yes

5. Review Comments to the Author

Reviewer #1: The paper provides statistics and analysis about the citations and co-citations of academic publications on Wikipedia, one of the largest encyclopedia available to humans for free. It reveals the significant presence of "Medicine” and “Biochemistry, Genetics, and Molecular Biology” papers, as well as the important role of multidisciplinary, high-impact factor journals. It also highlights that only 13.44% of the citations are Open Access.

The authors cleansed the dataset carefully, validating the citations on Wikipedia. This is critical as subsequent analysis all depends on the validation of the citations.

The results of this paper, despite its large amount, show little significance as findings and are sometimes misleading. The authors often abruptly stop their investigation after presenting some statistics, leaving the critical and interesting questions unanswered. Not only are the numbers hard to interpret correctly, but the paper also lacks a cohesive narrative. As a result, I have a difficult time answering the following question after reading this paper, "what have I learned from this paper"?

For example, the paper states that each Wikipedia entry includes on average 4.373 references. However, older and more popular entries may be better developed, and therefore includes more references. It is rather confusing what information or message this average delivers to the readers, especially when the standard error is so large as well (8.351). Since the distribution is likely skewed (judging from the large stderr and what's commonly observed in citation analysis), the averages may not representative statistics after all.

Is the distribution of references per entry/edit log-normal or power-law? On the per-entry level, how does this depend on the entry's age, birth year/month, popularity, topic, language, etc? On the per-edit level, how does this depend on the editors' experience, expertise area, edit year/month, language, etc? Can you show causality via matching? Are these relationships different when comparing editors on Wikipedia and researchers in academia? Stronger/weaker homophily in languages?

The authors compare Scopus citations with Wikipedia citations in mean absolute difference. The total number of citations is likely different in Scopus and Wikipedia. If Scopus has significantly more total citations in its system, it is rather unsurprising most paper has more citations in Scopus than in Wikipedia. The hypothesis here can be "are the distribution of citations for the papers in Scopus significantly different from that in Wikipedia?" If you can reject the hypothesis, that gives a motivation for investigating what factors drive the tested difference in the distribution.

Similarly, the finding that "General Medicine" and "Molecular Biology" stand out from other fields can be a result of the distribution of papers in each field. Imagine the editors are citing papers in random by throwing darts, trolling the system in some sense. If "General Medicine" and "Molecular Biology" produces significantly more papers, the random darts are more likely to land on papers in these two fields as well.

There needs to be more evidence to support the main findings of this paper, namely from the social perspective of Wikipedia, the authors find a significative presence of “Medicine” and “Biochemistry, Genetics, and Molecular Biology” papers, and that the most important journals are multidisciplinary. One may find the same result from any perspectives, not exclusive to the social perspective of Wikipedia. There may not be anything particular about Wikipedia that associates with this observation. One way is to clarify this doubt is to analyze the co-citation network of both Wikipedia and Scopus, and test if there is a significant difference in the distribution of fields and the ranking of journal importance between Wikipedia and Scopus.

I believe the dataset at the hands of the authors can reveal much more than a set of descriptive findings. In particular, the author can focus more on what is special about Wikipedia and its social aspect that makes a part of the presented observations interesting, and follow up with more in-depth analysis targeting those observations. However, at the moment I deem the paper insufficient for publication.

Reviewer #2: This is an interesting study analyzing the relationship

between scientific papers and Wikipedia. While it is well writen

in most of its parts, the following issues should be addressed

before it can be accepted:

1) This is study is directed related to the topic of creating scientific

maps, a well-known research topic in the scientometric field. The authors should

improve the section of related works, mentioning similar works. See and mention e.g. doi: 10.1016/j.joi.2016.03.008; doi: 10.1007/s11192-012-0784-8; The analysis of knowledge in information networks has also been conisdered in recent works:

doi: 10.1016/j.ins.2017.08.091

2) The authors should better explain why not using other types of networks to create the study. What is the effect of just using co-citation networks. Several options exists: citation networks, co-references, no use of references at all (e.g. text analysis). This possibility should be at least considered.

3) Why the authors used a dataset that has not been updated?

4) The authors should better explain the particular choice for a minimum covering tree and particular centrality measurements. The motivation for using such features of complex netowrks is not clear.

5) It is also not clear how interdisciplinarity (symmetry) is measured in graphs. It would be interesting

the authors to mention some of the possible approaches in the literature and how

such methods could obtain different results. See and mention e.g. the follwoing examples: entropic diversity (doi: 10.1209/0295-5075/110/68001); quantum walks (doi: 10.1103/PhysRevE.88.032806) and concentric rings.

Once the above major issues are addressed I will be able to recommend publication.

6. PLOS authors have the option to publish the peer review history of their article (what does this mean?). If published, this will include your full peer review and any attached files.

Reviewer #1: Yes: Pik-Mai Hui

Reviewer #2: No

---

## [Author Response · Author response to Decision Letter 0]

14 Dec 2019

Reviewer 1:

We thank the reviewer for the in-depth review. Considering the criticism made, we have carried out an in-depth revision of the aspects indicated (writing, data analysis, etc.), particularly those indicated in his/her comments. Below we detail the changes made and the modifications introduced in the text for each of the criticisms made by the reviewer.

The comments are interesting and we have followed the main recommendations that the reviewer points out. But it should be considered that the analysis and in-depth citation patterns in Wikipedia are not the aim of the paper. Our main objective is to make a representation of its structure through co-citation. Also, the reviewer indicates questions that are far from this objective and that therefore cannot be answered at this point because we did not collect this type of information or fields (i.e. topics, popularity, editor, etc. ...). In addition, in line with our initial objectives, we opted to use Altmetric.com as our data source as it provides the sort of information that we needed and no other, i.e.: Mention Date (date of incorporation of the reference), Mention Title (title of the entry) and Mention URL (link to the revision of the entry in which the reference is introduced). Therefore many of the analyses indicated cannot be performed.

However, we do consider that the suggestions made by the reviewer can contribute to and improve the interpretation of the maps. That is why, first of all, we have conducted a more exacting statistical analysis in the following areas: 1) analysis of the annual evolution of citations by entry and of their accumulated value; 2) statistics concerning co-citations; and 3) a study of the multidisciplinarity of references.

In relation to averages and standard deviations, the reviewer is right and we expanded our explanation of this phenomenon. On the one hand, we have analyzed the Wikipedia entries that include more references, detecting, for example, those related to annual lists of scientific discoveries in a given discipline. On the other hand, we have also explained that this variation is not unknown in metrics of this type, since the distribution curve presents a particularly marked asymmetry as 81% of articles receive only one citation; those that receive between one and three citations account for 97% of the total. These are also seen in the fact that 20% of the most cited works constitute only 40% of the total number of citations. This phenomenon also occurs in almost all bibliometric indicators.

In this sense, following the reviewer's recommendations, we have carried out an analysis of the distribution of citations received by journals, which fits both log-normal and power-law. This explanation can be found, both in the results section and, in greater detail, in a new section that we have included in the discussion (Comparison of Wikipedia and Scopus).

This comparison was weakly explained in our first version of the paper and, therefore, we have edited it following the reviewers’ recommendation. Our aim was to determine whether the most cited works in Wikipedia coincided with those in Scopus. But obviously the work is not intended to analyze the citations or their distribution. However, we have followed the reviewer's advice and, as indicated in the previous reply, we have analyzed the distribution of citations in Wikipedia using the journals and comparing it with the results achieved in other works. 

In addition to correcting this paragraph, we have exploited our data to improve the comparison between Wikipedia and Scopus citations at journal level in order to determine whether they offer different information. To do this we have used the number of citations included in the CiteScore Metrics (citations made in 2016 to articles published between 2013 and 2015) and adjusted those of Wikipedia to the same citation window. As we have commented in the text, the two statistical models generated offer an R2 of around 0.5 that we have not considered sufficient to demonstrate that there is strong relationship between both sets.

We thank the reviewer for pointing to the subject of trolling, which is one of the great problems of most social media, and we have incorporated the following paragraphs:

“The Wikipedia references, unlike those collected in other social media, offer remarkable quality control and transparency. In relation to the problem of trolling, the encyclopedia is based on a solid quality management system of post-publication peer review in which, in the case of discrepancies, changes are resolved through consensus between editors. Wikipedia also has two manuals: for non-academic experts (https://en.wikipedia.org/wiki/Help:Wikipedia_editing_for_non-academic_experts) and for researchers, scholars, and academics (https://en.wikipedia.org/wiki/Help:Wikipedia_editing_for_researchers,_scholars,_and_academics) both specifying the importance of the use of citations under the principles of verifiability and notability. This substantially minimizes the likelihood that references in entries will be tampered with.

Wikipedia also offers a complete list of its bots (https://en.wikipedia.org/wiki/Special:ListUsers/bot), including those such as the Citation bot (https://en.wikipedia.org/wiki/User:Citation_bot), which in addition to adding missing identifiers to references, corrects and completes them, something for which the digital encyclopedia offers several tools (https://en.wikipedia.org/wiki/Help:Citation_tools). However, this does not prevent the appearance of publications with a high, anomalous number of citations [42]. For instance, we found a report (https://www.altmetric.com/details/3171944) cited in 1450 lunar crater entries, not attributable to a bot. So, although the use of citations is not compromised, practices of this sort must be taken into account, for example, if their use is in an evaluative context. Given all of the aforementioned, we consider that in this context in which 193 802 Wikipedia entries and 847 512 article citations have been analyzed, it is very difficult to produce manipulations that could significantly alter the system and, consequently, the results achieved here.”

We agree with the reviewer as long as, in principle, it may seem that there are similarities in the predominance of certain scientific areas (Medicine, ...) and certain types of journals (multidisciplinary). Although the reviewer indicates that the results are not exclusive to Wikipedia, we do consider that there are singular results in Wikipedia that are a direct consequence of the social practices of the platform. In this regard, we must indicate the specific case of "Biochemistry", which has a better representation—more acute in Wikipedia than in Scopus. The same happens with other areas such as "Agricultural and Biological Science", as observed in Figure 8. However, we do agree with the reviewer that these have not been properly described in the text. We have therefore improved the results and the discussion. We have emphasized the singularities of Figure 8, written two new paragraphs in the discussion, and added a new table as complementary material, which clarifies this and helps to better interpret the co-citation maps. 

“Like other platforms, multidisciplinary journals are hubs and articulate the Wikipedia network. However, despite being a common global phenomenon, Wikipedia does have, and contains unique citation practices mentioning journals that are not cited or mentioned in a relevant way in other databases or platforms. This is evidenced in the scatter plot of Wikipedia and Scopus citations (see Fig 5).

This difference is also illustrated by a comparison of the journals most mentioned in Altmetric.com with those most mentioned in Wikipedia. As we can see, Wikipedia has the major multidisciplinary journals in common, as does Scopus. However, some of the most frequently mentioned journals in Wikipedia are the Journal of Biological Chemistry or Zookeys which are located in JCR’s Q2. Nonetheless, the Journal of Biological Chemistry, for example, is one of the most widely cited journals in the field of genetics receiving a large number of citations in various entries such as "Androgen receptor" (45 citations) or "Epidermal growth factor receptor" (25 citations). Therefore, Wikipedia points to another type of specialized journal in different fields (See Table S8) that are not identified in other rankings. In addition, as we have indicated, this can hardly be due to trolling or a bot.”

Reviewer 2:

We thank the reviewer for their suggestions. In addition to including these in our new text, we have improved our literature review by including nine more references:

● 1. Brzezinski M. Power laws in citation distributions: evidence from Scopus. Scientometrics [Internet]. 2015;103(1):213–28. Available from: https://doi.org/10.1007/s11192-014-1524-z

● 2. Chang Y-W, Huang M-H, Lin C-W. Evolution of research subjects in library and information science based on keyword, bibliographical coupling, and co-citation analyses. Scientometrics [Internet]. 2015;105(3):2071–87. Available from: https://doi.org/10.1007/s11192-015-1762-8

● 3. Costas R, de Rijcke S, Marres N. Beyond the dependencies of altmetrics: Conceptualizing ‘heterogeneous couplings’ between social media and science. In: The 2017 Altmetrics Workshop [Internet]. 2017. Available from: http://altmetrics.org/wp-content/uploads/2017/09/altmetrics17_paper_4.pdf

● 4. Didegah F, Thelwall M. Co-saved, co-tweeted, and co-cited networks. J Assoc Inf Sci Technol [Internet]. 2018 Aug 1;69(8):959–73. Available from: https://doi.org/10.1002/asi.24028

● 5. Seglen PO. Why the impact factor of journals should not be used for evaluating research. BMJ [Internet]. 1997 Feb 15;314(7079):498–502. Available from: https://www.ncbi.nlm.nih.gov/pubmed/9056804

● 6. Torres-Salinas D, Gorraiz J, Robinson-Garcia N. The insoluble problems of books: what does Altmetric.com have to offer? Aslib J Inf Manag. 2018;70(6):691–707. 

● 7. Trujillo CM, Long TM. Document co-citation analysis to enhance transdisciplinary research. Sci Adv [Internet]. 2018 Jan 1;4(1):e1701130. Available from: http://advances.sciencemag.org/content/4/1/e1701130.abstract

● 8. van Eck NJ, Waltman L. Visualizing Bibliometric Networks BT - Measuring Scholarly Impact: Methods and Practice. In: Ding Y, Rousseau R, Wolfram D, editors. Cham: Springer International Publishing; 2014. p. 285–320. Available from: https://doi.org/10.1007/978-3-319-10377-8_13

● 9. Yan E, Ding Y. Scholarly network similarities: How bibliographic coupling networks, citation networks, cocitation networks, topical networks, coauthorship networks, and coword networks relate to each other. J Am Soc Inf Sci Technol [Internet]. 2012 Jul 1;63(7):1313–26. Available from: https://doi.org/10.1002/asi.22680

The reviewer is right and in the methodology section we have included a new paragraph in which we explain and justify the selection of the co-citation networks.

“Co-citation networks, bibliographic coupling and direct citations are some of the most significant bibliometric networks we can use to map citations from Wikipedia entries; of these, co-citation networks are the most popular in research [29,30]. If we take into account other types of network such as co-author and co-word, the aforementioned three methods show a high degree of similarity [31]. Within the field of altmetrics, the concepts of co-citation and coupling have both been adapted [32], but co-citations offer more varied alternatives [33]. Furthermore, they are of special interest as they have been identified as capable of enhancing transdisciplinarity [34]. Hence, we have generated co-citation maps at the level of journal, field and main field.”

In our case, apart from the fact that the data used is one year old, when we started working on this study the only dataset available with the information we needed to carry it out was the CiteScore Metrics.Furthermore, until then Elsevier had updated this set annually, but then stopped giving it continuity.

We have updated the methodology to better explain the choice of the Pathfinder algorithm and have included in the results a comparison between the network with and without Pathfinder, also introducing a new figure.

“The Pathfinder algorithm [35] has been applied as a pruning method following a common configuration (r=∞, q=n-1) that reduces the networks to a minimum covering tree. This algorithm—successfully applied in the field of Library and Information Sciences [26, 36]—keeps only the strongest co-citation links between all pairs of nodes and offers a diaphanous view of large networks. Given the huge amount of co-citations, especially between journals, we use this technique to prune them in order to make the networks more explanatory.”

We appreciate the suggestion of the reviewer, but within the analysis of networks we have not analyzed the symmetry distribution. Our objective has been to map co-occurrences at different levels (journal, main field and field) established on the basis of citations from scientific articles. This is why we have decided to use the Pathfinder algorithm to highlight the strongest relationships.

---

## [Decision Letter · Decision Letter 1]

19 Dec 2019

PONE-D-19-25486R1

Science through Wikipedia: a novel representation of open knowledge through co-citation networks

PLOS ONE

Dear Dr. Torres-Salinas,

Thank you for submitting your manuscript to PLOS ONE. After careful consideration, we feel that it has merit but does not fully meet PLOS ONE’s publication criteria as it currently stands. Therefore, we invite you to submit a revised version of the manuscript that addresses the points raised during the review process.

We would appreciate receiving your revised manuscript by Feb 02 2020 11:59PM. To enhance the reproducibility of your results, we recommend that if applicable you deposit your laboratory protocols in protocols.io, where a protocol can be assigned its own identifier (DOI) such that it can be cited independently in the future. For instructions see: http://journals.plos.org/plosone/s/submission-guidelines#loc-laboratory-protocols

We look forward to receiving your revised manuscript.

Kind regards,

Sergi Lozano

Academic Editor

PLOS ONE

Additional Editor Comments (if provided):

Dear authors,

as you can see below, both reviewers are now more positive regarding the manuscript. In your revision of the manuscript, I would like you to pay especial attention to comments by Reviewer 1 on selection of the correlation test to apply (considering data skewness) and interpretation of results (including p-values).

Sergi.

Reviewers' comments:

Reviewer's Responses to Questions

**Comments to the Author**

1. If the authors have adequately addressed your comments raised in a previous round of review and you feel that this manuscript is now acceptable for publication, you may indicate that here to bypass the “Comments to the Author” section, enter your conflict of interest statement in the “Confidential to Editor” section, and submit your "Accept" recommendation.

Reviewer #1: All comments have been addressed

Reviewer #2: All comments have been addressed

2. Is the manuscript technically sound, and do the data support the conclusions?

Reviewer #1: Yes

Reviewer #2: Yes

3. Has the statistical analysis been performed appropriately and rigorously? 

Reviewer #1: No

Reviewer #2: N/A

4. Have the authors made all data underlying the findings in their manuscript fully available?

Reviewer #1: Yes

Reviewer #2: (No Response)

5. Is the manuscript presented in an intelligible fashion and written in standard English?

Reviewer #1: Yes

Reviewer #2: Yes

6. Review Comments to the Author

Reviewer #1: I appreciate the replies from the authors and their effort in enhancing the paper.

The paper, in my opinion, has mapped science through the lens of Wikipedia. The comparison between Wikipedia and Scopus then reveals how differently topics and journals represent among Wikipedia editors and among academic researchers. At the end the paper talks about specific findings.

With the additional statistical analysis, I believe the authors have done a good job mapping science on Wikipedia. However, I would like to suggest improvement about Table S1. Table S1 should include either mean/var of log-normal fit or exponent of power-law fit, with likelihood-ratio statistics of power-law model compared to log-normal model. Please be referred to

Clauset, Aaron, Cosma Rohilla Shalizi, and Mark EJ Newman. "Power-law distributions in empirical data." SIAM review 51.4 (2009): 661-703. pdf: https://arxiv.org/abs/0706.1062

for the rationales and methods for doing so. They also provide packages in R and Python.

This leads to suggestions about Figure 5. Since both distributions are skewed, R2 is not an appropriate. Pearson R assumes finite variances, which sometimes breaks down when variables are power-law distributed. While I believe the conclusion that the correlation is not large, it is important to use appropriate method to reach this conclusion in the paper. I suggest the following two approaches:

1. Use rank correlation, such as Spearman's Rho or Kendall's Tao.

2. Use QQ plot to support the finding of outliers. Quantiles (p-value) at each axis should be computed using the power-law/log-normal models fitted for Scopus citation distribution and Wikipedia citation distribution.

I believe suggestion 2 above can replace Figure 5, which is currently too disperse. But if Figure 5 needs to stay for any reason, it should be plotted in log-log scale. Be careful to not fit linear models in log-scale, because variances do not remain constant in log-scale. One should fit the models before log-transform, and then visualize the fitted lines in log-scale. Suggestion 2 avoids all these complications.

From suggestion 2, in fact, the authors can sort journals by the ratio of p-value of Wikipedia citation to that of Scopus under power-law models. The high-ratio journals are "over-cited" in Wikipedia, while the low-ratio journals are "under-cited", compared to Scopus as a baseline. I believe both the "over-cited" and "under-cited" ones worth some characterization and discussion. The authors may want to only look at journals with enough data to avoid noise in the ratio. Note that PLoS ONE and PNAS may not have the highest ratios, which suggests that the finding in the second paragraph of the result section may not be entirely correct.

This kind of comparison using ratio also applies in Figure 8. Looking at ratios, articles in the following fields gather more attentions in Wikipedia:

- Multidisciplinary

- Biochemistry, Genetics and Molecular Biology

- Agricultural and Biological Science

- Earth and Planetary Science

- Immunology and Microbiology

while the following fields are less popular in Wikipedia:

- Engineering

- Material Science

- Mathematics

- Chemistry

- Chemical Engineering

The less popular fields are all extremely hard, but steadily evolving sciences, which suggests that the limited ability of Wikipedia editors in comprehending articles in these fields restricted their ability to cite the articles. However, I have no empirical evidence nor reference to support this conjecture.

The field "Medicine" has roughly same percentages of attention in both Scopus and Wikipedia, despite the fact that it is the most popular field for both. This differs from the finding in the first paragraph of the result section.

As a side note, the authors also need to be careful when interpreting the correlation coefficients, which come with p-values. The size and significance of a correlation coefficient, while related, are not the same. The correlation can be small in size but statistically significant (e.g. huge sample and weak correlation), or large in size but not statistically significant (e.g. moderate correlation in a very small sample). For example, a correlation of 0.5, while statistically significant, means a sizable correlation. This correlation defines a pattern, then one can identify outlier points that do not obey this pattern.

Lastly, I would like to suggest the removal of the phrase "social perspective" throughout the paper. These is no analysis in this paper between the social aspect of Wikipedia editors and any of the finding. An example of such type of analysis can be the correlation of citation fields between editors who communicated in edit review, i.e. a social, communication network of editors. Merely co-edit of an entry does not imply any social interaction between the involved editors. While this paper is mapping science through the lens of Wikipedia, I have difficulty seeing how this mapping is done through social relations of editors.

After the first revision, I can see ithe structure of the paper and start to appreciate its contribution. Therefore I recommend a minor revision. After responding the comments above, I believe the paper is useful to other researchers and will make a significant impact in Wikipedia research.

Reviewer #2: All issues have been addressed, therefore I recommend acceptance for this particular manuscript. ====

7. PLOS authors have the option to publish the peer review history of their article (what does this mean?). If published, this will include your full peer review and any attached files.

Reviewer #1: Yes: Pik-Mai Hui

Reviewer #2: No

---

## [Author Response · Author response to Decision Letter 1]

25 Dec 2019

We appreciate the reviewer’s suggerence but we have not carried out our power-law analysis by fields as suggested to improve Table S1. We used journals’ citations to illustrate this phenomenon in Wikipedia without dividing them by areas or papers (each journal is one observation). However, the recommended R package is the same that we used in this analysis which now has been referenced.

29. Gillespie CS. Fitting Heavy Tailed Distributions: The poweRlaw Package. J Stat Softw [Internet]. 2015 Mar 20;64(2). Available from: http://doi.org/10.18637/jss.v064.i02

Following the reviewer’s recommendations about Figure 5 we have updated it with colors by the ratio of Wikipedia and Scopus citation percentiles, and also included Q-Q plot using the normal and log-normal models and Figure 5 in log-log scale as supplementary material. These two new figures have been obtained taking into account the reviewer’s considerations. Furthermore, comments for both figures are included in results and discussion.

We appreciate the recommendation to analyse the ratio of Wikipedia and Scopus citations under log-normal/power-law models. For this reason the over-cited and under-cited journals have been determined. To calculate them we filtered the dataset to only journals with a minimum of 2 papers and 3 Wikipedia and Scopus cites to avoid noise and then we calculated the ratio using their citation percentiles. As result we have included:

“In this sense we have obtained the journals’ citation percentiles in Wikipedia and Scopus, using only journals with a minimum of three citations in both platforms and two articles cited to avoid noise, and then the ratio between these percentiles have been calculated. While the commented journals have the same attention (ratio=1), the over-cited ones in Wikipedia are Mammalian Species(3559), Art Journal (192.56), Northern History (126.92), European Journal of Taxonomy (83.92) and Art Bulletin (80.92), and the under-cited ones are Physical Review A - Atomic, Molecular, and Optical Physics, Dalton Transactions and Applied Surface Science (all of them with 0.00027).”

The reviewer is right, the ratio analysis is useful to study the Wikipedia attention by fields. In this sense, the ratio of Wikipedia and Scopus citations would complete the results related to Figure 8. However, we have the limitation of Scopus citations, that only cover some years and papers, so we can not compare these results to Figure 8 properly.

In relation with the correlation coefficient comments we have reviewed them. Furthermore, these explications have been completed with the comments included about outliers.

We have changed “social perspective” to “Wikipedia perspective”.

---

## [Decision Letter · Decision Letter 2]

23 Jan 2020

Science through Wikipedia: a novel representation of open knowledge through co-citation networks

PONE-D-19-25486R2

Dear Dr. Torres-Salinas,

We are pleased to inform you that your manuscript has been judged scientifically suitable for publication and will be formally accepted for publication once it complies with all outstanding technical requirements.

With kind regards,

Sergi Lozano

Academic Editor

PLOS ONE

Additional Editor Comments (optional):

Reviewers' comments:

Reviewer's Responses to Questions

**Comments to the Author**

1. If the authors have adequately addressed your comments raised in a previous round of review and you feel that this manuscript is now acceptable for publication, you may indicate that here to bypass the “Comments to the Author” section, enter your conflict of interest statement in the “Confidential to Editor” section, and submit your "Accept" recommendation.

Reviewer #1: All comments have been addressed

2. Is the manuscript technically sound, and do the data support the conclusions?

Reviewer #1: Yes

3. Has the statistical analysis been performed appropriately and rigorously? 

Reviewer #1: Yes

4. Have the authors made all data underlying the findings in their manuscript fully available?

Reviewer #1: Yes

5. Is the manuscript presented in an intelligible fashion and written in standard English?

Reviewer #1: Yes

6. Review Comments to the Author

Reviewer #1: All comments have been addressed. I appreciate the authors' detailed response to my comments. I recommend acceptance as is.

7. PLOS authors have the option to publish the peer review history of their article (what does this mean?). If published, this will include your full peer review and any attached files.

Reviewer #1: Yes: Pik-Mai Hui

---

## [Editor Report · Acceptance letter]

30 Jan 2020

PONE-D-19-25486R2 

Science through Wikipedia: a novel representation of open knowledge through co-citation networks 

Dear Dr. Torres-Salinas:

I am pleased to inform you that your manuscript has been deemed suitable for publication in PLOS ONE. Congratulations! Your manuscript is now with our production department. 

With kind regards,

on behalf of

Dr. Sergi Lozano 

Academic Editor

PLOS ONE